# Safety and Efficacy of Embolic Protection Devices in Saphenous Vein Graft Interventions: A Propensity Score Analysis—Multicenter SVG PCI PROTECTA Study

**DOI:** 10.3390/jcm9041198

**Published:** 2020-04-22

**Authors:** Wojciech Wańha, Maksymilian Mielczarek, Natasza Gilis-Malinowska, Tomasz Roleder, Marek Milewski, Szymon Ładziński, Dariusz Ciećwierz, Paweł Gąsior, Tomasz Pawłowski, Rafał Januszek, Adam Kowalówka, Michalina Kolodziejczak, Stanisław Bartuś, Marcin Gruchała, Grzegorz Smolka, Eliano Pio Navarese, Dariusz Dudek, Andrzej Ochała, Elvin Kedhi, Miłosz Jaguszewski, Wojciech Wojakowski

**Affiliations:** 1Department of Cardiology and Structural Heart Diseases, Medical University of Silesia, 40-635 Katowice, Poland; marek.milewski92@gmail.com (M.M.); szymon.ladzinski@gmail.com (S.Ł.); p.m.gasior@gmail.com (P.G.); tomaszzpawlowski@gmail.com (T.P.); gsmolka@me.com (G.S.); aochala1@gmail.com (A.O.); ekedhi@yahoo.com (E.K.);; 2First Department of Cardiology, Medical University of Gdansk, 80-210 Gdańsk, Poland; max.mielczarek@gmail.com (M.M.); tasza.gilis@gmail.com (N.G.-M.); cedario@o2.pl (D.C.); mgruch@gumed.edu.pl (M.G.); jamilosz@gmail.com (M.J.); 3Regional Specialist Hospital, Research, and Development Center, 51-124 Wroclaw, Poland; tomaszroleder@gmail.com; 4Second Department of Cardiology, Jagiellonian University Medical College, 31-501 Krakow, Poland; jaanraf@interia.pl (R.J.); mbbartus@gmail.com (S.B.); mcdudek@cyf-kr.edu.pl (D.D.); 5Department of Cardiac Surgery, Medical University of Silesia, 40-635 Katowice, Poland; adam.kowalowka@orange.pl; 6Department of Anaesthesiology and Intensive Care, Antoni Jurasz University Hospital No. 1, 85-094 Bydgoszcz, Poland; kolodziejczak.michalina@gmail.com; 7SIRIO MEDICINE research network, 85-094 Bydgoszcz, Poland; elianonavarese@gmail.com; 8Cardiovascular Institute, Nicolaus Copernicus University, 85-094 Bydgoszcz, Poland; 9Department of Cardiovascular Medicine, University of Alberta, Edmonton, AB 13-103, Canada; 10AZ Sint Jan, 8000 Brugge, Belgium

**Keywords:** percutaneous coronary intervention, saphenous vein graft, embolic protection devices

## Abstract

*Background*: Evidence concerning the efficacy of the embolic protection devices (EPDs) in saphenous vein graft (SVG) percutaneous coronary intervention (PCI) is sparse. The study was designed to compare major cardiovascular events of all-comer population of SVG PCI with and without EPDs at one year of follow-up. *Methods and results*: A multi-center registry comparing PCI with and without EPDs in consecutive patients undergoing PCI of SVG. The group comprised 792 patients, among which 266 (33.6%) had myocardial infarction (MI). The primary composite endpoint was major adverse cardiac and cerebrovascular event (MACCE) defined as death, MI, target vessel revascularization (TVR), and stroke assessed at one year. After propensity score analysis, there were no differences in MACCE (21.9% vs. 23.9%; HR 0.91, 95% CI 0.57–1.45, *p* = 0.681, respectively) nor in secondary endpoints of death, MI, TVR, target lesion revascularization (TLR) and stroke at one year in EPDs PCI group vs. no-EPDs PCI group. Similarly, there were no differences between groups in the study endpoints at 30 days follow-up. *Conclusions*: There were no clinical benefit for routine use of EPDs during SVG PCI in short and long-term follow-up. Further studies are warranted to explore the effect of individual types of EPDs on clinical outcomes.

## 1. Introduction

Saphenous vein graft (SVG) failure affects 15–35% of patients within five years and 29–68% of patients ten years after the procedure [1,2]. Vein graft stenosis is characterized by an accelerated and different progression of atherosclerosis as compared with the process occurring in native coronary arteries [3]. Plaque morphology in SVG consist of friable tissue prone to cause distal embolization during percutaneous coronary intervention (PCI) [4,5]. Thus, the PCI of SVG, in comparison to native vessels, is associated with an increased risk of peri-procedural complications, predominantly myocardial infarction or the no-reflow [6]. Therefore, the rationale behind the use of embolic protection devices (EPDs) is to reduce the risk of distal embolization and thus improves SVG PCI outcomes. Currently, the application rate of EPDs during SVG PCI ranges between 7% and 22% [7]. In recent years, stent technology, procedural techniques, and drug therapy showed substantial progress, which resulted in excellent native coronary PCI outcomes [8,9]. However, that is not so obvious in SVG PCI, including the controversies with regards to use of EPDs during SVG PCI. The results of several single-center studies evaluating clinical endpoints demonstrated benefits of EPDs, that were, however, not corroborated by other SVG PCI registries with no significant improvement when compared with no-EPDs [10,11]. Conflicting results from new studies translated to the recent amendment in European Society of Cardiology (ESC) guidelines [12], which gave a class II a recommendation to use EPDs in SVG intervention (a downgrade from class I given in the previous recommendations) [13]. Nevertheless, in patients with degenerated SVG, an adjunct use of EPDs might still contribute to reduction of distal embolization and preservation of myocardial perfusion. Therefore, in SVG PCI PROTECTA Study we decided to carry out an all-comer registry with short and long-term follow-up of patients with significant SVG stenosis treated with PCI and with or without EPD.

## 2. Methods

The study flow chart is presented in the Figure 1. The SVG Baltic Registry served as a data source for this multicenter study [14]. The study was performed in three, high volume PCI centers between 2008 and 2014. The retrospective registry included all consecutive patients with significant SVG stenosis treated with PCI. Importantly, patients who had PCI of other vascular territories during the same procedure were excluded. The inclusion and exclusion criteria and information about data collection was published previously are listed in the SVG Baltic Registry [14]. Baseline clinical characteristics of patients and procedural data were collected and recorded in the central registry. Outcome data were obtained from the National Health Fund Service central database of the Ministry of Health. All patients completed one year of follow -up. In case of re-PCI in the follow up we additionally checked TVR and TLR. The patient’s datasets were anonymized in each center, combined into one, and analyzed as a single cohort. The patient’s data were protected in accordance with the requirements of Polish law and hospital Standard Operating Procedures (SOPs). A total of 792 patients who received isolated SVG PCI were included. Use of EPDs types was as follows: Spider FXTM (Medtronic, Minneapolis, MN, USA) (50%), FilterWire EZ (Boston Scientific, Natick, MA, USA) (29%), EmboShield^®^ (Abbott Laboratories, Santa Clara, CA, USA) (8%), Defender (Medtronic, Minneapolis, MN, USA) (7%), RX Accunet (Abbott Vascular, Santa Clara, CA, USA) (3%), Proxis (St Jude Medical, St Paul, MN, USA) (3%). To elucidate patients who could benefit more distinctly from EPDs, a subanalysis for those with myocardial infarction (MI) was performed.

### 2.1. End-Points

We studied major adverse cardiac and cardiovascular events (MACCE, a composite of all cause death, MI, target vessel revascularization—TVR and stroke) and single components of MACCE at 30-days and 1-year follow-up. The primary efficacy endpoint was MACCE at one year. The secondary endpoints were MACCE at 30-days as well as all-cause death, MI, TVR, target lesion revascularization (TLR) and stroke at 30-days and 1-year follow-up. Due to observational character of the study the propensity score matching was performed. Propensity score matching was applied for all endpoints at one year follow-up. Study outcomes were defined in accordance to the universal definitions of clinical trials endpoints [15].

### 2.2. Statistical Analysis

Continuous data were presented as mean ± standard deviation or median with interquartile range (Q1–Q3). Qualitative data were expressed as crude values and/or percentages. Normal distribution was verified by the Kolmogorov–Smirnov test. Continuous data were compared by Student *t*-test or by U Mann–Whitney test depending on the distribution. Categorical data were analyzed by the Chi-square test and Fisher’s exact test. Kaplan–Meier survival curves were performed to present the unadjusted time-to-event data for investigated end-points and were calculated using the log-rank test. To limit biases, propensity score matching analysis was used. The logistic regression was performed with EPDs as a dependent variable, and following independent variables: age, gender, length of hospitalization, anemia, chronic renal failure, hypertension, diabetes, smoking, dyslipidaemia, prior MI, prior PCI, family history of CAD, peripheral artery disease (PAD), carotid artery disease, previous neoplasms, lung disease, STEMI on admission, vessel access, previous stent thrombosis, periprocedural MI, in-stent restenosis, total stent length, average stent diameter, use of predilatation, residual stenosis, postprocedural TIMI-3, no-reflow, dissection and number of stent implanted. The validity of logistic regression was assessed using the Hosmer–Lemeshow test. The model was well calibrated (the Hosmer–Lemeshow test ×2 4.02, 8 df, *p* = 0.855). Then propensity score matching was performed using nearest neighbor methods—two groups of 155 patients each were created. Finally, Cox-regression for one-year event rates of MACCE, death, MI, TVR, TLR, and stroke was calculated for unmatched population and matched groups; *p*-value < 0.05 was considered significant. The statistical analysis was performed using Medcalc 17.9.2 (Medcalc software) and SPSS software v.21 (IBM SPSS Statistics).

## 3. Results

The Registry comprised consecutive patients that underwent PCI of isolated SVG stenosis, 266 of which (33.6%) presented with MI. The median age was 69 years, and the majority (602, 76.0%) of the studied patients were males. EPDs were used in 190 (23.9%) patients. Comparing EPDs vs. no-EPDs groups in a total study cohort there were no significant differences in patient’s baseline characteristics, and clinical presentation, except for smoking (*p* = 0.002) and family history of coronary artery disease (CAD) (*p* = 0.012). The groups were also comparable with regard to concomitant diseases (including diabetes, cancer, chronic obstructive pulmonary disease, peripheral and carotid artery disease). However, the median graft age was greater in EPDs vs. no-EPDs patients 13.8, IQR 10.7–16.4 vs. 12.7, IQR 8.8–15.6, *p* = 0.005) (Table 1).

### 3.1. In-Hospital and Discharge Medications

Glycoprotein IIb/IIIa inhibitors were used at the same rate in patients with and without EPDs (13.2% vs. 14.3%, *p* = 0.721), similarly to in-hospital use of oral antiplatelet and antithrombotic medications. Dual antiplatelet therapy (DAPT) was given for one month in all patients treated with bare-metal stent (BMS) in stable CAD or for 12 months in MI. In all patients treated with drug-eluting stent (DES), the DAPT was continued up to 12 months.

### 3.2. Interventional Treatment and Reperfusion Strategy

The most common native vessel territory receiving SVG PCI with EPDs was the left circumflex coronary artery. There were some differences in the use of EPDs, depending on the graft segment stenosis. In the mid segment stenosis, PCI were done more often with EPDs (21.6% vs. 14.8%, *p* = 0.027), whereas in the distal segment stenosis EPDs were used less often (8.9% vs. 24.6%, *p* < 0.001) (Table 2). The degree of stenosis and the presence of thrombus were comparable in both groups. EPDs were used more often in the new-DES group in comparison to the BMS group (58.9% vs. 44.4%, *p* < 0.001). While stent diameters were larger in SVG PCI with EPDs than those without EPDs (3.5, IQR 3.3–4.0 vs. 3.0, IQR 3.0–3.5, *p* < 0.001), stent lengths were similar. No differences in procedural success rate between SVG PCI with or without EPDs were observed (Table 2).

### 3.3. 1-Year and 30-Days Outcomes in the Allcommer Population

The primary endpoint of the study (1-year overall MACCE) was observed in 20.5% of patients in the EPDs PCI group, and in 26.4% of patients in the no-EPDs PCI group (HR 0.75, 95% CI 0.53–1.06, *p* = 0.105, Table 3 and Figure 2). This results remained not significant after propensity score analysis (21.9% vs. 23.9%; HR 0.91, 95% CI 0.57–1.45, *p* = 0.681, respectively).

At 1-year, there were no differences in the secondary endpoints: death (4.2% vs. 7.8% HR 0.53, 95% CI 0.25–1.12, *p* = 0.094), MI (7.4% vs. 10%; HR 0.73, 95% CI 0.41–1.30, *p* = 0.281), TVR (8.9% vs. 12.3%, HR 0.72, 95% CI 0.43–1.22, *p* = 0.220), TLR (6.8% vs. 8.8%; HR 0.78, 95% CI 0.43–1.43, *p* = 0.424), and stroke (1.1% vs. 1.7%; HR 0.63, 95% CI 0.14–2.89), *p* = 0.551) (Table 3 and Figure 3). There were no differences in those results after propensity analysis (Table 4). The subanalysis with EPDs vs. no-EPDs in the new-DES, and no-thrombectomy group did not reveal variations in rate of clinical outcomes at one year of follow up. Stent diameter also did not influence the overall results (Appendix A).

At 30-days follow up, there were no differences in MACCE in EPDs PCI group vs. no-EPDs PCI group (4.2% vs. 5.3% HR 0.79, 95% CI 0.36–1.71, *p* = 0.545, respectively) nor in the other secondary endpoints such as death (1.1% vs. 1.8% HR 0.90, 95% CI 0.19–4.33, *p* = 0.895), MI (1.1% vs. 1.3% HR 0.79, 95% CI 0.17–3.72, *p* = 0.765), TVR (1.6% vs. 1.0% HR 1.60, 95% CI 0.40–6.38, *p* = 0.509), TLR (1.6% vs. 0.5% HR 3.20, 95% CI 0.65–15.83, *p* = 0.155) and stroke (0.5 % vs. 0.7% HR 0.79, 95% CI 0.9–7.1, *p* = 0.833).

### 3.4. 1-Year Outcomes of the MI Group

In the MI study subpopulation, at 1-year follow-up, there were no differences in MACCE in EPDs PCI vs. no-EPDs PCI groups (24.6% vs. 34.6% HR 0.68, 95% CI 0.39–1.19, *p* = 0.174, respectively) nor in the secondary endpoints.

## 4. Discussion

The main finding of SVG PCI PROTECTA study was that during SVG PCI the use of EPDs was associated with comparable results to no-EPDs in all-comers population at 1-year follow-up, also after propensity score group matching.

The adjunct use of EPDs during SVG PCI is a matter of unceasing debate. Studies evaluating EPDs in SVG PCI demonstrated equivocal results concerning MACCE. In a large registry, SVG PCI with EPDs was associated with even higher incidence of procedural complications including no-reflow, vessel dissection, perforation, and peri-procedural MI, without corresponding improvements in clinical outcomes for up to three years afterwards, as compared to no-EPDs SVG PCI [16]. The significantly higher incidence of peri-procedural MI in EPDs SVG PCI was also noted in comprehensive review and meta-analysis prepared by Paul et al. [10]. The Randomized, Controlled Trial of SVG PCI with the TRAP-EPD (Microvena, White Bear Lake, MN) demonstrated no differences in deaths, MI, TVR, MACE in the TRAP-EPDs group compared with no-EPDs group at 30-days follow up [17]. These abovementioned outcomes challenged the role of adjunct EPDs in SVG PCI, questioning both its safety and efficacy. On the other hand, other studies showed favorable TIMI Grade Flow [18], reduction of no-reflow phenomenon, and lower 30-days MACCE when EPDs were used during SVG PCI [19]. Of note, a single-center study reported less prevalent MACE rate in patients with EPDs compared to no-EPDs at one year of follow-up [20]. We believe that conflicting study results highlight the need to elucidate the specific group of patients who would benefit from the adjunct use of EPDs. A number of factors might contribute to the characteristics of the most vulnerable SVG PCI patients’ group. Comorbidities, including diabetes mellitus, hypertension, dyslipidemia, not optimal pharmacological management or smoking, contribute to the baseline residual risk that affects all vessels, including native coronary arteries and SVG. The atherosclerotic plaque buildup in SVG varies from the one occurring in arteries. The progression is more rapid and atherosclerotic occlusions may occur even after 12–18 months from the index procedure [21]. The facilitation of the process occurs since venous wall is not adapted to arterial pressure load, which results in plaque formation of an atypical structure, with thinner, more friable fibrous caps. The revascularization procedures need to be more complex and delicate due to an increased risk of plaque embolization and platelet aggregation, as compared with native coronary artery lesions. If labile atheroembolic debris is liberated during an intervention, it might result in vasospasm, followed by slow or no-reflow phenomenon and subsequent periprocedural myocardial ischemia. In this context, the proper patient selection, revascularization technique, and EPDs mandates short- and long term protection of myocardium [21]. Because of this, patients with MI due to SVG thrombotic stenosis constitute the population that would most likely benefit from EPDs, even though such devices failed to prove their efficacy in patients with MI from native coronary vessels. Furthermore, it seems reasonable that different EPDs might not perform equally good in SVG environment. The devices vary in construction with regard to pore size, capture efficiency and the design of filter systems, either fixed on the end of the wire or mobile, which affects the continuity of flow. Operator must be vary of small particles, limitations of capture efficiency and procedural complications related to filter maneuvers in a particularly vulnerable vessel. Some EPDs allow for advancement without affecting blood flow and deployment once the vulnerable lesion was passed. Their efficacy, however, was not yet evaluated in a sufficiently powered study. Therefore, there is an unmet need to identify EPDs preferable in SVG PCI. These observations constructed a rationale for the SVG PCI PROTECTA study. The results of our study goes in line with previous large observational trials, which showed that the adjunct use of EPDs during SVG PCI was safe and did not increase periprocedural complications. The rate of deaths in EPDs group in our study was comparable with the results of the British Columbia Cardiac Registry, in which it occurred in 5.2% of patients at 12-months follow up [18]. Nevertheless, we did not find a benefit of adjunct use of EPDs during SVG PCI in all-comers population, but also in MI study subpopulation. It is reported that the major determinant for the EPDs use is operator’s preference [22]. It seems that more detailed studies accounting for device factors, clinical factors, and operator’s experience need to be conducted to identify populations of the highest benefit for a specific EPDs use in this environment

### Study Limitations

There are several limitations to this study. Lack of data on valvular heart disease, right heart failure and pulmonary hypertension. Lack of data on operator experience gained with a specific device. EPDs group patients received significantly more new-DES than BMS, therefore we additionally performed propensity score to limit above biases. Although the sample size was large, the study was not designed as a randomized trial, but a retrospective registry that has inherent limitations. However, this was balanced by “all-comer” population with no selective inclusion criteria, 100% follow-up rate and confirmation of the end-points by the National Health Service database, as well as robust statistical analyses, including propensity score analysis.

## 5. Conclusions

In patients undergoing PCI for treatment of SVG, the use of EPDs was not associated with any significant improvement in MACCE in an all-comers population both at 30-days and one year follow-up. Our results corroborate the current class IIa ESC guidelines recommendations on EPDs use in SVG PCI.

## Figures and Tables

**Figure 1 jcm-09-01198-f001:**
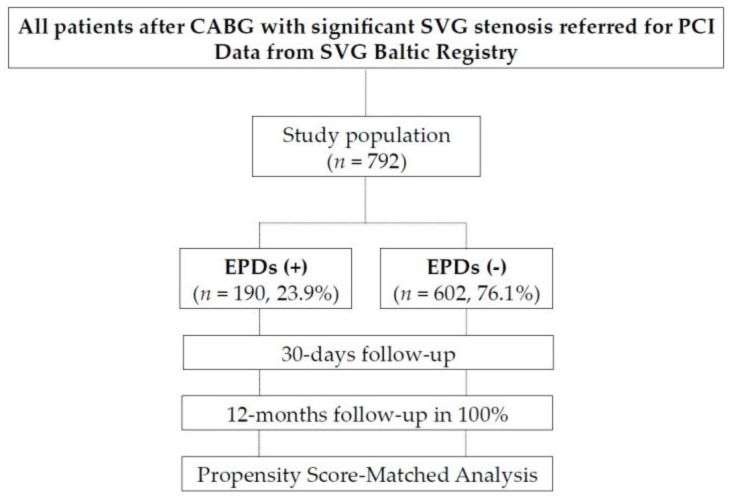
Study flow chart in the SVG PCI PROTECTA Study. CABG—coronary artery bypass graft, SVG—saphenous vein graft, PCI—percutaneous coronary intervention, EPDs—embolic protection devices.

**Figure 2 jcm-09-01198-f002:**
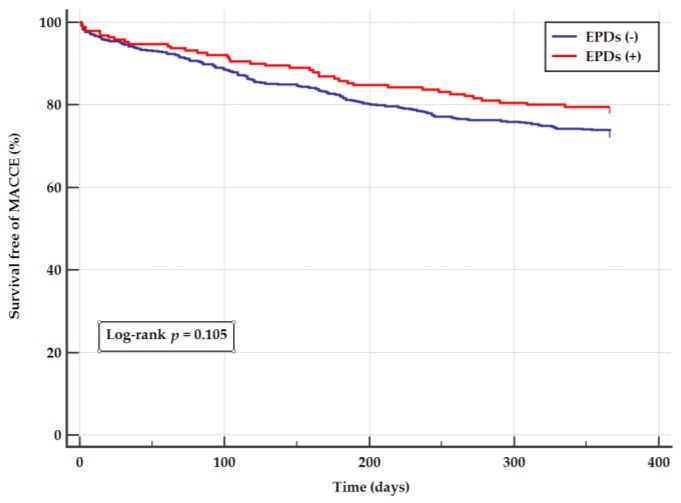
Kaplan–Meier Curves for MACCE (myocardial infarction, stroke, death, TVR) according to the use of embolic protection devices. MACCE—major adverse cardiac and cerebrovascular events.

**Figure 3 jcm-09-01198-f003:**
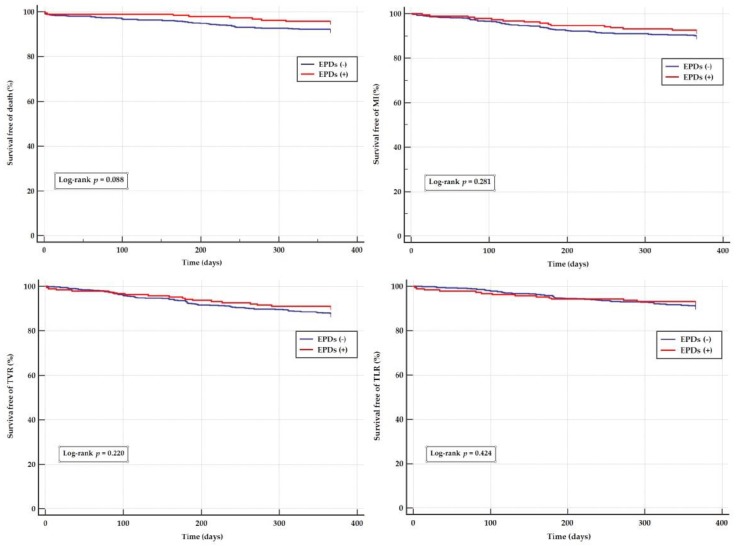
Kaplan–Meier Curves for death, myocardial infarction, TVR, TLR, according to the use of embolic protection devices. EPDs—embolic protection devices, MI—myocardial infarction, TLR—target lesion revascularization, TVR—target vessel revascularization.

**Table 1 jcm-09-01198-t001:** Patients characteristics, risk factors and clinical presentation according to the use of embolic protection devices.

	EPDs (+) *n* = 190 (23.9%)	EPDs (−) *n* = 602 (76.1%)	*p*
Demographic data			
Age, median (IQR)	70.0 (63.0–76.0)	69.0 (62.0–75.0)	0.124
Male, *n* (%)	145 (76.3)	457(75.9)	0.909
BMI (kg/m^2^), median (IQR)	29.0 (26.4–31.8)	27.8 (25.4–30.8)	0.111
Discharge diagnosis			
SA, *n* (%)	61 (32.1)	173 (28.7)	0.375
UA, *n* (%)	68 (35.8)	224 (37.2)	0.724
NSTEMI, *n* (%)	50 (26.3)	166 (27.6)	0.734
STEMI, *n* (%)	11 (5.8)	39 (6.5)	0.865
CAD history			
Previous MI, *n* (%)	137 (72.1)	422 (70.1)	0.597
Previous PCI, *n* (%)	93 (48.9)	324 (53.8)	0.241
Graft age, median (IQR)	13.8 (10.7–16.4)	12.7(8.8–15.6)	0.005
CAD risk factors			
Hypertension, *n* (%)	171 (90.0)	533 (88.5)	0.576
Dyslipidemia, *n* (%)	146 (76.8)	435 (72.3)	0.213
CKD, *n* (%)	42 (22.1)	159 (26.4)	0.234
Diabetes mellitus, *n* (%)	82 (43.2)	253 (42.0)	0.783
Current Smoking, *n* (%)	54 (28.4)	108 (17.9)	0.002
Family history of CAD, *n* (%)	67 (35.3)	156 (25.9)	0.012
Concomitant disease			
Cancer, *n* (%)	11 (5.8)	26 (4.3)	0.431
COPD, *n* (%)	14(7.3)	41(6.8)	0.792
PAD, *n* (%)	37 (19.5)	123 (20.4)	0.774
Carotid artery disease, *n* (%)	17 (8.9)	81 (13.5)	0.129
Length of hospital stay, median (IQR)	5.0 (3.0–7.0)	4.0 (3.0–6.0)	0.012
CCS, median (IQR)	3.0(2.0–4.0)	3.0(2.0–4.0)	0.939
NYHA, median (IQR)	1.0(1.0–2.0)	1.0(1.0–2.0)	0.097
LVEF, median (IQR)	50.0 (38.0–55.0)	48.0 (38.0–55.0)	0.427
GRACE score >140, *n* (%)	12 (15.2)	31 (14.2)	0.853
euroSCORE II, median, (IQR)	4.5 (3.0–9.0)	4.6 (2.8–8.2)	0.711

BMI—body mass index; BMS—bare metal stent, CABG—coronary artery bypass graft, CAD—coronary artery disease, CKD—chronic kidney disease (CKD was defined as estimated GFR (eGFR) <60 mL/min/1.73m^2^ calculated using the Modification of Diet in Renal Disease (MDRD) method), COPD—chronic obstructive pulmonary disease, DES—drug eluting stent, EPDs—embolic protection devices, MI—myocardial infarction, new-DES—new generation DES, NSTEMI—non-ST-segment elevation myocardial infarction, SA—stable angina, STEMI—ST-segment elevation myocardial infarction, PAD—peripheral artery disease, PCI—percutaneous coronary intervention, UA—unstable angina.

**Table 2 jcm-09-01198-t002:** Angiographic and procedural data according to the use of embolic protection devices.

	EPDs (+) *n* = 190 (23.9%)	EPDs (−) *n* = 602 (76.1%)	*p*
Access, *n* (%)			
Radial	27 (14.2)	84 (14.0)	0.955
Femoral	163 (85.8)	514 (86.0)	
Degree of stenosis, (%), median (IQR)	90.0 (80.0–95.0)	90.0 (80.0–95.0)	0.528
Thrombus, *n* (%)	19 (10.0)	47 (7.8)	0.366
Restenosis in previously implanted stent, *n* (%)	11 (5.8)	59 (9.8)	0.106
Stent thrombosis in previous implanted stent, *n* (%)	4 (2.1)	5 (0.8)	0.230
PCI, *n* (%)			
Ao-LAD	53 (27.9)	144 (23.9)	0.269
Ao-LCx	85 (44.7)	273 (45.3)	0.883
Ao-RCA	53 (27.9)	198 (32.9)	0.197
Ao-Y	8 (4.2)	30 (5.0)	0.846
Segment, *n* (%)			
Proximal	68 (35.8)	179 (29.7)	0.116
Mid	41 (21.6)	89 (14.8)	0.027
Distal	17 (8.9)	148 (24.6)	<0.001
Other	66 (34.7)	200 (33.2)	0.700
Stent, *n* (%)			
new-DES	112 (58.9)	267 (44.4)	<0.001
BMS	78 (41.1)	335 (55.6)	<0.001
Predilatation, *n* (%)	56 (29.5)	172 (28.6)	0.811
Thrombectomy, *n* (%)	10 (5.4)	32 (5.6)	0.902
Total stent length (mm), median (IQR)	20.0 (15.5–31.0)	20.0 (15.0–29.0)	0.083
Length of stent > 28mm, *n* (%)	64 (34.6)	177 (29.6)	0.198
Average stent diameter (mm), median (IQR)	3.5 (3.3–4.0)	3.0 (3.0–3.5)	<0.001
Stent diameter > 3.5mm, *n* (%)	140 (73.7)	284 (47.4)	<0.001
Number of implanted stents, median (IQR)	1.0 (1.0–2.0)	1.0 (1.0–1.8)	0.708
Residual stenosis, *n* (%)	5 (2.6)	19 (3.2)	0.813
TIMI 3 post-PCI, *n* (%)	187 (98.4)	592 (98.3)	0.938
Vessel perforation, *n* (%)	1 (0.5)	1 (0.2)	0.422
Dissection, *n* (%)	4 (2.1)	7 (1.2)	0.306
No reflow during PCI, *n* (%)	5 (2.6)	8 (1.3)	0.218
Periprocedural MI, *n* (%)	3 (1.6)	12 (2.0)	1.000
IIb/IIIa inhibitors, *n* (%)	25 (13.2)	86 (14.3)	0.721
Cardiac arrest, *n* (%)	1 (0.5)	10 (1.7)	0.475
Intra-aortic balloon pump, *n* (%)	0 (0)	10 (1.7)	-
Acute Stent thrombosis, *n* (%)	2 (1.1)	3 (0.5)	0.599
Subacute Stent thrombosis, *n* (%)	0 (0)	0 (0)	-
Calcifications, *n* (%)	3 (1.6)	3 (0.5)	0.279

BMS—bare metal stent, DES—drug eluting stent, EPDs—embolic protection devices, LAD—left anterior descending artery, LCx—left circumflex artery, MI—myocardial infarction, new-DES—new generation DES, PCI—percutaneous coronary intervention, RCA—right coronary artery, SVG—saphenous vein graft, TIMI—thrombolysis in myocardial infarction.

**Table 3 jcm-09-01198-t003:** One year follow up according to the use of embolic protection devices.

	EPDs (+) *n* = 190 (23.9%)	EPDs (−) *n* = 602 (76.1%)	Crude Calculation	
			HR (95% CI)	*p*
MACCE	39 (20.5)	159 (26.4)	0.75 (0.53–1.06)	0.105
Death	8 (4.2)	47 (7.8)	0.53 (0.25–1.12)	0.094
MI	14 (7.4)	60 (10.0)	0.73 (0.41–1.30)	0.281
Stroke	2 (1.1)	10 (1.7)	0.63 (0.14–2.89)	0.551
TVR	17 (8.9)	74 (12.3)	0.72 (0.43–1.22)	0.220
TLR	13 (6.8)	53 (8.8)	0.78 (0.43–1.43)	0.424

CI—confidence interval, EPDs—embolic protection devices, HR—hazard ratio, MACCE—major adverse cardiac and cerebrovascular events, MI—myocardial infarction, new-DES—new generation DES, TLR—target lesion revascularization, TVR—target vessel revascularization.

**Table 4 jcm-09-01198-t004:** One year follow up according to the use of embolic protection devices after propensity score matching.

	EPDs (+) *n* = 155	EPDs (−) *n* = 155	Propensity Score	
			HR (95% CI)	*p*-Value
MACCE	34 (21.9)	37 (23.9)	0.91 (0.57–1.45)	0.681
Death	5 (3.2)	9 (5.8)	0.54 (0.18–1.62)	0.271
MI	13 (8.4)	13 (8.4)	1.00 (0.46–2.15)	0.994
Stroke	1 (0.6)	1 (0.6)	1.00 (0.06–16.0)	0.998
TVR	16 (10.3)	20 (12.9)	0.80 (0.41–1.54)	0.497
TLR	12 (7.7)	16 (10.3)	0.75 (0.36–1.59)	0.451

CI—confidence interval, EPDs—embolic protection devices, HR—hazard ratio, MACCE—major adverse cardiac and cerebrovascular events, MI—myocardial infarction, new-DES—new generation DES, TLR—target lesion revascularization, TVR—target vessel revascularization.

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
