# Peer review of "Safety and Efficacy of Embolic Protection Devices in Saphenous Vein Graft Interventions: A Propensity Score Analysis—Multicenter SVG PCI PROTECTA Study"

_jcm, 2020, doi:10.3390/jcm9041198_

Round 1

Reviewer 1 Report

The paper entitled " Safety and Efficacy of Embolic Protection Devices in  Saphenous Vein Graft Interventions: A propensity  score analysis - Multicenter SVG PCI PROTECTA Study " retrospectively investigated the efficacy of EPDs in saphenous vein graft percutaneous coronary interventions.  The study is generally interesting but there are several major limitations that should be addressed.

Major:

  • A predisposition for the use of specific devices is found. This finding merits further discussion. Also, the device that may be more efficient was the one most commonly used. This is not surprising and may suggest that the more experience gained with a specific device – the more efficient it will be. Please elaborate the discussion accordingly.
  • Data collection was performed until 2014. Since then, many methods and pharmacotherapies were improved and the results may not be applicable to current practice. Please expand the cohort to include also patients treated during the years 2014-2020, who received newer antiaggregants (that should also be specified). 
  • Older DES were excluded while BMS were not, mostly since a previously generated database (not specifically designed for the purpose of the current study) was used. The inclusion of patients who received BMS seems inappropriate. I suggest that the study will include only patients treated with the new generation DES (class IA in the current ESC recommendations).
  • It is not entirely clear if patients underwent concomitant procedures to other vascular territories (other than the left or right internal mammary artery that was specified as exclusion criteria).
  • The definition of "stroke" is unclear. Were the patients evaluated by a neurologist? Underwent cranial CT scans? MRIs?
  • Euroscore is an outdated tool. Please specify euroSCORE II results (used since 2011). Also, data used in this scoring system (NYHA, CCS etc) should be specified.
  • Clinical data is missing. For example – did the patients have valvular problems? Right heart failure? Pulmonary hypertension?
  • " stent diameters were larger in SVG PCI with EPDs "  - this is a major possible confounder. Combined with the fact that the EPDs group received significantly more DES, the results may be jeopardized despite statistical corrections. Please address.
  • No reflow during PCI was more common in the EPDs group (2.6 vs 1.3%). Although it did not reach statistical significance, I expect that an EPD would be associated with lower degree of no-reflow (as this is the whole point of using it in the first place). Please discuss.
  • Regarding the Spider-EPDs group, a very wide range of 95%CI was reported on. This finding merits further discussion. Also, the characteristics of patients who did benefit from EPDs (compared to those who did not) should be specified. I suggest an analysis should be performed for all groups in order to compare those who benefited from those who did not. Also, a separate analysis for the 3 most commonly used EPDs can be of importance. This information may shed light on specific populations who may benefit from EPDs.
  • A sub-analysis excluding thrombectomy cases should be reported on (since this procedure may predispose to specific complications and may become a confounder).
  • A direct comparison between different EPDs should be conducted.

Minor:

  • The figures are of low resolution and should be re-plotted.
  • Please have the study registered in the clinicaltrial.gov database.
  • Line 213 " Paul et al. 17. " - the citation style should be kept throughout the manuscript.
  • The limitation section should be enhanced.
  • Timing of CABG should be specified.

Author Response

Reviewer 1.

Major:

1. A predisposition for the use of specific devices is found. This finding merits further discussion. Also, the device that may be more efficient was the one most commonly used. This is not surprising and may suggest that the more experience gained with a specific device – the more efficient it will be. Please elaborate the discussion accordingly.

Response: We thank the reviewer for the opportunity to address this point. The current study was performed in high volume PCI centres, which allowed to reach the highest standards of study conduction and minimized the risk of bias related related to the learning curve, where operators were experienced and trained with a variety of devices. The used device is widely used and largely reflects curent clinical practice. On the other hand, as the reviewer mentions, the personal experience and preference of a specific device could still affect the efficiency of one device over the other. The skills and experience of the operators are unfortunately a common issue in procedural studies, even of a randomized design, where a margin of a decision-making is left upon operator’s preference and centre’s device availability. As per recommendation, we now elaborate on this issue in the discussion and additionally decided to list in the limitations paragraph. (Discussion, Study Limitations Lines 250-252 ; 254-257).

2. Data collection was performed until 2014. Since then, many methods and pharmacotherapies were improved and the results may not be applicable to current practice. Please expand the cohort to include also patients treated during the years 2014-2020, who received newer antiaggregants (that should also be specified). 

Response: The current SVG PCI PROTECTA study is a part of the SVG Baltic Registry, with outcome data obtained from the central database of the National Health Fund Service of the Ministry of Health which has been designed with the highest methodological standards of data collection with no reported losses at patient follow-up. The SVG Baltic Registry was conducted between February 2008 and October 2014 and once the individual centres datasets were completed, they were anonymized, combined into one database, and analyzed as a single cohort. Taking into account that original registry and the main study analysis was already completed it would not be feasible to collect the data of patients from 2014-2020 period for a substudy only.

3. Older DES were excluded while BMS were not, mostly since a previously generated database (not specifically designed for the purpose of the current study) was used. The inclusion of patients who received BMS seems inappropriate. I suggest that the study will include only patients treated with the new generation DES (class IA in the current ESC recommendations).

Response: We thank the reviewer for the possibility to clarify this point. As we state in the submitted manuscript, the SVG PCI PROTECTA study is a part of the SVG Baltic Registry, however, this was not the reason for exclusion of the older generation DES. In the present study we aimed to present the most contemporary real-life data on SVG revascularization. In this sense, the study is timely in offering clinicals insights into the PCI in SVG with more modern DESs. Including the old-DES would be contra intuitive, since they are no longer available on the market. At the same time, BMS not only are still in clinical use, particularly in patients at the high risk of bleeding1 that require short DAPT to prevent from bleeding complications, but also are evaluated hand-in-hand with new DES in randomized clinical trials in SVG revascularization2.

Additionally, in our registry, we previously assessed the impact EDPs in DES subpopulation. Nevertheless, we did not observe any differences in any components of MACCE (Table 1 - supplement material).

1) Shah R, Rao SV, Latham SB, et al. Efficacy and safety of drug-eluting stents optimized for biocompatibility vs bare-metal stents with a single month of dual antiplatelet therapy: a meta-analysis. JAMA Cardiol. 2018;Epub ahead of print.

2) Brilakis ES, Edson R, Bhatt DL, et al. Drug-eluting stents versus bare-metal stents in saphenous vein grafts: a double-blind, randomised trial. Lancet. 2018;391(10134):1997–2007.

Table 1. (Supplement material)

1-year follow up according to the use of EPDs in DES subanalysis.

EDPs (+)

n=112

EDPs (-)

n=267

Crude Calculation

HR (95% CI)

p-value

MACCE

19 (17.0)

62 (23.2)

0.68 (0.38-1.19)

0.177

Death

5 (4.5)

21 (7.9)

0.55 (0.20-1.49)

0.238

MI

5 (4.5)

19 (7.1)

0.61 (0.22-1.68)

0.338

Stroke

0 (0.0)

5 (1.9)

-

0.999

TVR

8 (7.1)

34 (12.7)

0.53 (0.24-1.18)

0.119

TLR

6 (5.4)

21 (7.9)

0.66 (0.26-1.69)

0.389

CI: confidence interval, EPDs- embolic protection devices, HR- hazard ratio, MACCE- major adverse cardiac and cerebrovascular events, MI- myocardial infarction, TLR- target lesion revascularization, TVR- target vessel revascularization

4. It is not entirely clear if patients underwent concomitant procedures to other vascular territories (other than the left or right internal mammary artery that was specified as exclusion criteria).

Response: We now have clarified the text to follow Reviewer’s suggestions: “Importantly, patients who had PCI of other vascular territories during the same procedure were excluded.” (Methods, Lines 69-70).

5. The definition of "stroke" is unclear. Were the patients evaluated by a neurologist? Underwent cranial CT scans? MRIs?

Response: The outcome data (stroke included) were obtained from the central database of the Polish National Health Fund Service of the Ministry of Health, and were classified according to universal criteria of ICD-10. The outcome was classified as “stroke” when patient was hospitalized and treated for the event at the designed stroke department - a full evaluation by a neurologist and at minimum CT scan was therefore mandatory to make a decision upon the treatment strategy. (Methods, Lines 72-74).

6. Euroscore is an outdated tool. Please specify euroSCORE II results (used since 2011). Also, data used in this scoring system (NYHA, CCS etc) should be specified.

Response: We thank the reviewer for this insightful point. There was a typo that we have now corrected. We have indeed calculated euroSCORE II. Additional we add NYHA and CCS in the table (Table 1).

7. Clinical data is missing. For example - did the patients have valvular problems? Right heart failure? Pulmonary hypertension?

Response: We thank the reviewer for his comments . The design of SVG PCI PROTECTA study reflected the current evidence base. Therefore, since other major registries1,2 did not report valvular or structural abnormalities, we also did not collect the abovementioned data.

1 Brennan, J. M.; Al-Hejily, W.; Dai, D.; Shaw, R. E.; Trilesskaya, M.; Rao, S. V.; Brilakis, E. S.; Anstrom, K. J.; Messenger, J. C.; Peterson, E. D.; Douglas, P. S.; Sketch, M. H., Jr., Three-year outcomes associated with embolic protection in saphenous vein graft intervention: results in 49 325 senior patients in the Medicare-linked National Cardiovascular Data Registry CathPCI Registry. Circulation. Cardiovascular interventions 2015, 8 (3), e001403.

2 Iqbal, M. B.; Nadra, I. J.; Ding, L.; Fung, A.; Aymong, E.; Chan, A. W.; Hodge, S.; Della Siega, A.; Robinson, S. D.; British Columbia Cardiac Registry, I., Embolic protection device use and its association with procedural safety and long-term outcomes following saphenous vein graft intervention: An analysis from the British Columbia Cardiac registry. Catheterization and cardiovascular interventions : official journal of the Society for Cardiac Angiography & Interventions 2016, 88 (1), 73-83. 

8." stent diameters were larger in SVG PCI with EPDs "  - this is a major possible confounder. Combined with the fact that the EPDs group received significantly more DES, the results may be jeopardized despite statistical corrections. Please address.

Response: We thank the reviewer for this important comment. In our registry, we already assessed the impact of the stent diameter on the clinical endpoints at one year of follow-up. Nevertheless, we did not observe any differences in any components of MACCE (Table 3 - supplement material).

As the reviewer mentioned, in the EPDs group patients received significantly more DES than BMS and the results of SVG Baltic registry (from the same database) showed that in patients undergoing PCI of SVG, the use of new-DES is associated with a reduced 1-year rate of MACCE and MI compared to BMS. Therefore, in SVG PROTECTA study we additionally performed propensity score to further match the investigated population based also on the type of DES devices implanted . We have clarified it in the study results and limitations to follow Reviewer’s suggestions.

Table 3. (Supplement material)

Impact of the stent diameter on the one year follow up.

Univariate logistic regression method was used.

HR

95%CI

p

MACCE

0.86

0.66-1.13

0.275

Death

0.83

0.52-1.33

0.443

MI

1.16

0.79-1.72

0.450

Stroke

1.06

0.41-2.70

0.908

TVR

0.80

0.55-1.16

0.237

TLR

0.83

0.54-1.27

0.390

HR- hazard ratio, MACCE- major adverse cardiac and cerebrovascular events, MI- myocardial infarction, TLR- target lesion revascularization, TVR- target vessel revascularization,

9. No reflow during PCI was more common in the EPDs group (2.6 vs 1.3%). Although it did not reach statistical significance, I expect that an EPD would be associated with lower degree of no-reflow (as this is the whole point of using it in the first place). Please discuss.

Response: We thank the reviewer for this important comment. EPDs despite being a protective device by design, can cause procedural complications with embolization of plaque debris when passed through a lesion, in particular including no-reflow. Moreover, previous studies1 reported even more procedural complications with EPDs as compared to no-EPDs in SVG PCI, including: no-reflow, vessel dissection and perforation. Thus, the most delicate device should be used for PCI SVG. To further address this point we have clarified this phenomenon in the discussion to follow Reviewer’s suggestions.

1 Brennan, J. M.; Al-Hejily, W.; Dai, D.; Shaw, R. E.; Trilesskaya, M.; Rao, S. V.; Brilakis, E. S.; Anstrom, K. J.; Messenger, J. C.; Peterson, E. D.; Douglas, P. S.; Sketch, M. H., Jr., Three-year outcomes associated with embolic protection in saphenous vein graft intervention: results in 49 325 senior patients in the Medicare-linked National Cardiovascular Data Registry CathPCI Registry. Circulation. Cardiovascular interventions 2015, 8 (3), e001403.

10. Regarding the Spider-EPDs group, a very wide range of 95%CI was reported on. This finding merits further discussion. Also, the characteristics of patients who did benefit from EPDs (compared to those who did not) should be specified. I suggest an analysis should be performed for all groups in order to compare those who benefited from those who did not. Also, a separate analysis for the 3 most commonly used EPDs can be of importance. This information may shed light on specific populations who may benefit from EPDs.

Response: We thank the reviewer for this important comment. A wide range of 95%CI was is due to the small number of endpoints in the spider-EPDs group, which warrants further studies in larger study group. In order to comply with all points raised by reviewers we decided to remove the Spider-EPDs group analysis as per second reviewer’s multiple suggestions. We would like to refer the first reviewer to the respectable section of the rebuttal for full rationale of the amendments applied to our manuscript.

11. A sub-analysis excluding thrombectomy cases should be reported on (since this procedure may predispose to specific complications and may become a confounder).

Response: We have now performed a sub-analysis excluding thrombectomy – we did not observe any differences in all components of MACCE (Table 2 - supplement material).

Table 2. (Supplement material)

1-year follow up according to the use of EPDs in the no-thrombectomy sub-group.

EDPs (+)

n=180

EDPs (-)

n=570

Crude Calculation

HR (95% CI)

p-value

MACCE

36 (20.0)

147 (25.8)

0.72 (0.48-1.09)

0.116

Death

7 (3.9)

43 (7.5)

0.50 (0.22-1.12)

0.092

MI

13 (7.2)

55 (9.6)

0.73 (0.39-1.37)

0.325

Stroke

2 (1.1)

10 (1.8)

0.63 (0.14-2.90)

0.552

TVR

16 (8.9)

70 (12.3)

0.70 (0.40-1.23)

0.697

TLR

12 (6.7)

50 (8.8)

0.74 (0.39-1.43)

0.743

 CI: confidence interval, EPDs- embolic protection devices, HR- hazard ratio, MACCE- major adverse cardiac and cerebrovascular events, MI- myocardial infarction, TLR- target lesion revascularization, TVR- target vessel revascularization

12. A direct comparison between different EPDs should be conducted.

Response: To comply with points listed by all reviewers we would like to refer the first reviewer to 3rd point raised by the second reviewer: “I would not perform subgroup analysis based on types of EPDs as the sample size is really small (total EPD 190)”. We therefore decided not to perform analyses between different EPDs as it would decrease numbers of patients per group and ultimately provide unreliable estimates.

Minor:

1. The figures are of low resolution and should be re-plotted.

Response: Thank you, we increased the resolution of the figures.

2. Please have the study registered in the clinicaltrial.gov database.

Response: Thank you, we have we have sent a application to https://register.clinicaltrials.gov

3. Line 213 " Paul et al. 17. " - the citation style should be kept throughout the manuscript.

Response: Thank you, we corrected this, accordingly.

4. The limitation section should be enhanced.

Response: We updated the limitation section as per reviewer’s suggestions.

5. Timing of CABG should be specified.

Response: Thank you, we have added timing of CABG, accordingly (Table 1).

Reviewer 2 Report

Wańha W et al evaluated the Safety and Efficacy of Embolic Protection Devices in Saphenous Vein Graft Interventions with a propensity score analysis in a Multicenter SVG PCI Study. This study gives further evidence in the role of EPD in SVG PCI with previous conflicting results and guideline recommendation was derived from only one RCT published in 2002.
This study has clinical significance and has impact in our day to day clinical practice.

I have following comments:

Major comments:

1. I recommend to change this sentence: “however, a subanalysis comparing SVG PCI with Spider-EPDs vs. no-EPDs groups, presented a trend towards lower risk of death in the device group (2.1% vs. 7.8% HR 0.26, 95%CI 0.06-1.07, p=0.062, respectively)”.

The p value is 0.062, So, I strongly recommend to state that “additionally there is no difference in death using Spider-EPDs vs. no-EPDs groups, (2.1% vs. 7.8% HR 0.26, 95%CI 0.06-1.07, p=0.062). I will remove this from both abstract result and conclusion and from main manuscript results and conclusion.

2. I recommend to delete these 2 sentences from the Highlight section due to above reason:
SVG PCI using Spider-EPDs showed favourable clinical outcomes in terms of 1-year rate of death compared to no-EPDs.
• SVG PCI PROTECTA Study support Spider-EPDs as the mainstay EPDs technology in patients undergoing PCI of SVG.

3. I would not perform subgroup analysis based on types of EPDs as the sample size is really small (total EPD 190), also in USA only filter EZ is being used.

Would delete “To elucidate patients who could benefit more distinctly from EPDs, a subanalysis for those with myocardial infarction (MI) and for those with Spider-EPD guided PCI was performed. The Spider-EPDs is the EPDs with an exceptional construction as opposed to other EPD devices. The mobile nitonol mesh filter is delivered
over the guidewire and deployed for use. Spider-EPDs is additionally equipped with a gold loop around the mouth of the filter for verification of full apposition
before proceeding with the procedure”

4. 30 day and 1 year outcomes:
I recommend to revise “there was a trend towards a lower risk of death in EPDs PCI group vs. no-EPDs PCI group (4.2% vs. 7.8% HR 0.53, 95%CI 0.25-1.12, p= 0.094)”. Rewrite that there was no difference as P = 0.094.

5. In the same token, I will revise this sentence: The trend towards lower mortality at 1-year in EPDs PCI vs no-EPDs PCI groups was lost after propensity score analysis (3.2% vs. 5.8%; HR 0.54, 95%CI 0.18-1.62, p=0.271)”
Will write there was no difference in this result after propensity match analysis

6. I strongly recommend to delete this sentence:
However, a trend towards lower risk of death in Spider-EPDs group was noted (2.1% vs. 7.8% HR 0.26, 95%CI 0.06-1.07, p=0.062).
7. Will delete these sentences:
The results of our study have exciting implications. It showed particularly promising results in the subgroup of Spider-EPDs. Based on these results, it may be speculated that the specific technology of Spider-EPDs supports its use in SVG environment over the other EPDs.
Will rewrite no difference in Spider and no-EPD

8. Please delete “The results were more promising when Spider-EPDs were used, which warrants further studies” from conclusions.

9. The Randomized, Controlled Trial of SVG PCI with a TRAP-EPDs
(Microvena, White Bear Lake, MN) demonstrated a trend towards a lower
incidence of MI in the TRAP group compared with no-EPDs in 30-days follow up.
(19). It appears to me that this is a wrong interpretation on MI and why you would only discuss MI here.

Please double check the above statement: TRAP trial showed no benefit as below:

MACE at 30-day TRAP device vs Control 17.3% vs 12.7 % (p =0.24)
No difference in all-cause mortality, MI and TVR at 30 days.

Minor comments:

There are several grammatical error, especially singular and pleural , recommend to review the manuscript carefully.

Please elaborate in the abstract: MI, MACCE

Introduction:
“but, other clinical trials have shown no significant improvement when compared with SVG PCI without EPDs (10-12). Will change clinical trials to studies. And remove reference 10 from here as this study showed benefit with EPD.

Cardiology (ESC) guidelines (13)- use which gives

Methods
Use of EPDs types was as follows: use were as follows

Please Elaborate all abbreviations in table 1. and table 2: SA, LBBB and so forth

End points:
adverse cardiac and cardiovascular events: major adverse cardiac and cerebrovascular evets ( not cardiac and cardiovascular events).Please use one time elaboration and then consistently use abbreviated term , do not need both all the time.

Statistical analysis:
The logistic regression was performed with EPDs as a dependent variable, including age etc – this is an incomplete sentence meaning other variables as covariates? Please separate the sentence or rewrite.

It would be better if you can describe the timeframe of data collection year to year

What do you men aby medial segment? Do you mean mid segment?
Instead of writing EPD PCI and no EPD PCI group I would write EPD vs no -EPD group as it is redundant

Discussion:
that most likely might benefit most from EPDs” rewrite this sentence: most likely benefit from EPDs

Author Response

Reviewer 2.

Wańha W et al evaluated the Safety and Efficacy of Embolic Protection Devices in Saphenous Vein Graft Interventions with a propensity score analysis in a Multicenter SVG PCI Study. This study gives further evidence in the role of EPD in SVG PCI with previous conflicting results and guideline recommendation was derived from only one RCT published in 2002. This study has clinical significance and has impact in our day to day clinical practice.

I have following comments:

Major comments:

1. I recommend to change this sentence: “however, a subanalysis comparing SVG PCI with Spider-EPDs vs. no-EPDs groups, presented a trend towards lower risk of death in the device group (2.1% vs. 7.8% HR 0.26, 95%CI 0.06-1.07, p=0.062, respectively)”. The p value is 0.062, So, I strongly recommend to state that “additionally there is no difference in death using Spider-EPDs vs. no-EPDs groups, (2.1% vs. 7.8% HR 0.26, 95%CI 0.06-1.07, p=0.062). I will remove this from both abstract result and conclusion and from main manuscript results and conclusion.

Response: Thank you, we corrected the manuscript and removed the abovementioned sentences, accordingly.

2. I recommend to delete these 2 sentences from the Highlight section due to above reason: SVG PCI using Spider-EPDs showed favourable clinical outcomes in terms of 1-year rate of death compared to no-EPDs. SVG PCI PROTECTA Study support Spider-EPDs as the mainstay EPDs technology in patients undergoing PCI of SVG.

Response: Thank you, we corrected this, accordingly.

3. I would not perform subgroup analysis based on types of EPDs as the sample size is really small (total EPD 190), also in USA only filter EZ is being used.

Response: To comply with points listed by all reviewers we would like to refer the first reviewer to 12th point raised by the first reviewer: “A direct comparison between different EPDs should be conducted”. We therefore decided not to perform analyses between different EPDs as it would decrease numbers of patients per group and ultimately provide unreliable estimates.

4. Would delete “To elucidate patients who could benefit more distinctly from EPDs, a subanalysis for those with myocardial infarction (MI) and for those with Spider-EPD guided PCI was performed. The Spider-EPDs is the EPDs with an exceptional construction as opposed to other EPD devices. The mobile nitonol mesh filter is deliveredover the guidewire and deployed for use. Spider-EPDs is additionally equipped with a gold loop around the mouth of the filter for verification of full apposition before proceeding with the procedure”

Response: Thank you, we have now removed the said lines from the manuscript.

5. 30 day and 1 year outcomes: I recommend to revise “there was a trend towards a lower risk of death in EPDs PCI group vs. no-EPDs PCI group (4.2% vs. 7.8% HR 0.53, 95%CI 0.25-1.12, p= 0.094)”. Rewrite that there was no difference as P = 0.094.

Response: Thank you, we corrected this, accordingly.

6. In the same token, I will revise this sentence: The trend towards lower mortality at 1-year in EPDs PCI vs no-EPDs PCI groups was lost after propensity score analysis (3.2% vs. 5.8%; HR 0.54, 95%CI 0.18-1.62, p=0.271)” 
Will write there was no difference in this result after propensity match analysis

Response: We thank the reviewer for this important comment. We agree with Reviewer and have changed the text accordingly.

7. I strongly recommend to delete this sentence: However, a trend towards lower risk of death in Spider-EPDs group was noted (2.1% vs. 7.8% HR 0.26, 95%CI 0.06-1.07, p=0.062).

Response: Thank you, we removed the lines, as per reviewer’s suggestion.

8. Will delete these sentences: The results of our study have exciting implications. It showed particularly promising results in the subgroup of Spider-EPDs. Based on these results, it may be speculated that the specific technology of Spider-EPDs supports its use in SVG environment over the other EPDs. Will rewrite no difference in Spider and no-EPD

Response: Thank you, we corrected the manuscript and removed the abovementioned sentences.

9. Please delete “The results were more promising when Spider-EPDs were used, which warrants further studies” from conclusions.

Response: Thank you, we deleted the lines.

10. The Randomized, Controlled Trial of SVG PCI with a TRAP-EPDs (Microvena, White Bear Lake, MN) demonstrated a trend towards a lower incidence of MI in the TRAP group compared with no-EPDs in 30-days follow up.(19). It appears to me that this is a wrong interpretation on MI and why you would only discuss MI here.

Response: Thank you, we corrected this, accordingly.

11. Please double check the above statement: TRAP trial showed no benefit as below: MACE at 30-day TRAP device vs Control 17.3% vs 12.7 % (p =0.24). No difference in all-cause mortality, MI and TVR at 30 days.

Response: Thank you, we corrected this, accordingly.

Minor comments:

1. There are several grammatical error, especially singular and pleural , recommend to review the manuscript carefully.

Response: Thank you, we reviewed the manuscript for language corrections.

2. Please elaborate in the abstract: MI, MACCE

Response: Thank you, we now provide details on the abbreviations MI and MACCE.

Introduction: 

3. “but, other clinical trials have shown no significant improvement when compared with SVG PCI without EPDs (10-12). Will change clinical trials to studies. And remove reference 10 from here as this study showed benefit with EPD.

Response: Thank you, we corrected this, accordingly.

4. Cardiology (ESC) guidelines (13)- use which gives

Response: Thank you, we corrected this, accordingly.

Methods

5. Use of EPDs types was as follows: use were as follows

Response: Thank you, we corrected this, accordingly.

6. Please Elaborate all abbreviations in table 1. and table 2: SA, LBBB and so forth

Response: Thank you, we now provide details on the abbreviations.

7. End points: adverse cardiac and cardiovascular events: major adverse cardiac and cerebrovascular evets (not cardiac and cardiovascular events).Please use one time elaboration and then consistently use abbreviated term , do not need both all the time.

Response: Thank you, we corrected this, accordingly.

8. Statistical analysis: The logistic regression was performed with EPDs as a dependent variable, including age etc – this is an incomplete sentence meaning other variables as covariates? Please separate the sentence or rewrite.

Response: Thank you, we corrected this, accordingly.

9. It would be better if you can describe the timeframe of data collection year to year

Response: Thank you, we described the timeframe of data collection, accordingly.

10. What do you men aby medial segment? Do you mean mid segment? 
Instead of writing EPD PCI and no EPD PCI group I would write EPD vs no -EPD group as it is redundant

Response: Thank you, precise term is mid segment, we amended it throughout the manuscript.

11. Discussion: that most likely might benefit most from EPDs” rewrite this sentence: most likely benefit from EPDs

Response: Thank you, we corrected this, accordingly.

Round 2

Reviewer 2 Report

All comments were answered appropriately.